# Reconfigurable Metalens with Phase-Change Switching between Beam Acceleration and Rotation for 3D Depth Imaging

**DOI:** 10.3390/mi13040607

**Published:** 2022-04-13

**Authors:** Zhiyuan Ma, Siyu Dong, Xiong Dun, Zeyong Wei, Zhanshan Wang, Xinbin Cheng

**Affiliations:** 1MOE Key Laboratory of Advanced Micro-Structured Materials, Shanghai 200092, China; 1851924@tongji.edu.cn (Z.M.); dunx@tongji.edu.cn (X.D.); weizeyong@tongji.edu.cn (Z.W.); wangzs@tongji.edu.cn (Z.W.); chengxb@tongji.edu.cn (X.C.); 2Institute of Precision Optical Engineering, School of Physics Science and Engineering, Tongji University, Shanghai 200092, China; 3Shanghai Frontiers Science Center of Digital Optics, Shanghai 200092, China; 4Shanghai Institute of Intelligent Science and Technology, Tongji University, Shanghai 200092, China

**Keywords:** metalens, depth imaging, reconfigurable metasurface, point spread function

## Abstract

Depth imaging is very important for many emerging technologies, such as artificial intelligence, driverless vehicles and facial recognition. However, all these applications demand compact and low-power systems that are beyond the capabilities of most state-of-art depth cameras. Recently, metasurface-based depth imaging that exploits point spread function (PSF) engineering has been demonstrated to be miniaturized and single shot without requiring active illumination or multiple viewpoint exposures. A pair of spatially adjacent metalenses with an extended depth-of-field (EDOF) PSF and a depth-sensitive double-helix PSF (DH-PSF) were used, using the former metalens to reconstruct clear images of each depth and the latter to accurately estimate depth. However, due to these two metalenses being non-coaxial, parallax in capturing scenes is inevitable, which would limit the depth precision and field of view. In this work, a bifunctional reconfigurable metalens for 3D depth imaging was proposed by dynamically switching between EDOF-PSF and DH-PSF. Specifically, a polarization-independent metalens working at 1550 nm with a compact 1 mm^2^ aperture was realized, which can generate a focused accelerating beam and a focused rotating beam at the phase transition of crystalline and amorphous Ge_2_Sb_2_Te_5_ (GST), respectively. Combined with the deconvolution algorithm, we demonstrated the good capabilities of scene reconstruction and depth imaging using a theoretical simulation and achieved a depth measurement error of only 3.42%.

## 1. Introduction

The emergence of optical imaging has greatly enriched our lives and enhanced the human ability to perceive and reform the world. However, traditional cameras can only display two-dimensional plane projections of real 3D scenes without any depth information. With the rapid development of many emerging fields, deep imaging has attracted extensive attention in numerous vital applications, such as obstacle avoidance [1], product quality detection [2], driverless vehicles [3] and face recognition [4]. So far, there have been many approaches to obtain depth information, for example, the optical time-of-flight (TOF) [5,6], lens array [7], structured light projection [8], and depth from defocus [9]. However, all of these methods either need active illumination, multiple viewpoint exposures, or dynamic scanning, all of which greatly increase the system volume and power consumption.

Some distinctive PSFs that are different from those of the traditional lenses can also help to obtain the depth information of objects like DH-PSF [10,11]. DH-PSF lenses can generate two foci in the focal plane that continuously rotate in response to the change in object depth [11,12,13]. However, the sidelobe energy in DH-PSF will limit the reconstructed image quality and depth precision [14]. The combination of a DH-PSF lens and an EDOF lens, whose PSF is highly invariant to depth, has been proposed to achieve clear depth images [15], because EDOF lenses can help to reconstruct clear images of each depth with a single-shot capture, and can provide the additional reference images for a DH-PSF lens. However, this method was usually realized in aid of the spatial light modulators by switching the phase mask between EDOF and DH-PSF lenses [16,17,18,19], which required more optical elements to manipulate wavefront and increased the system volume and complexity.

Metasurfaces [20,21] that manipulate electromagnetic waves by artificial subwavelength scatterers [21,22] bring us the hope for more complex light-field regulation [23,24,25] and miniaturized optical systems. For metasurface-based depth imaging, many strategies in traditional methods can be directly extended to design a metasystem. For example, the metalens array has been used for depth imaging [26], but its imaging resolution is limited by the small aperture of the unit metalens [26,27]. The defocus method has also been investigated using a bifocal metalens [28], but its depth precision is limited by the intrinsic depth of field of the metalens [11,15]. Another work designed a metalens with a large chromatic focal shift to acquire the depth information, but this method only accommodates for a narrow depth detection range [29]. Alternatively, on the basis of PSF engineering in Ref. [15], Colburn et al. proposed utilizing two spatially adjacent metalenses with a depth-invariant EDOF-PSF and a depth-sensitive DH-PSF, respectively, to achieve 3D depth imaging [30]. This method can enable a compact and snapshot depth camera; however, due to the two metalenses being non-coaxial, parallax in capturing scenes is inevitable, which would limit the depth precision and field of view.

In recent years, research on the active metasurface [31], whose optical properties can be effectively controlled by an external electric field [32,33], light [34], or heat [35], so as to realize the function conversion that cannot be achieved by traditional passive metasurface, has made rapid progress. To overcome the non-coaxial disadvantages in Ref. [30], here we propose a bifunctional reconfigurable metalens for 3D depth imaging by dynamically switching between EDOF and DH-PSF metalenses based on the phase change material GST. Two independent phase profiles that generate a focused accelerating beam and a focused rotating beam, respectively, were designed in the crystalline (C-state) and amorphous state (A-state) GST at 1550 nm with a 1 mm^2^ aperture. Because of the nanosecond-level conversion speed of GST, our metalens can provide the 3D depth imaging capabilities with almost a single snapshot in completely coaxial imaging. Through the theoretical simulation, we have shown the clear reconstructed depth images of scenes with both a single object and multiple objects using our method, with only a 3.42% depth measurement error.

## 2. Results

### 2.1. Design Principle of the Bifunctional Metalens

Our bifunctional metalens is a co-aperture device that features two different and complementary PSFs obtained in the A-state and C-state GST, respectively. This enables simultaneous scene reconstruction and depth acquisition. As shown in Figure 1a, the metalens generates two blurred images on the sensor successively by phase-change conversion. The two images obtained are then processed with the deconvolution algorithm to produce both a reconstructed clear image and a corresponding depth map. Due to the flexible wavefront manipulation and integration characteristics of the metasurfaces, we can integrate the functions of an imaging lens and the additional functions of wavefront coding into a single surface by setting the phase profiles of the metalens as the sum of a standard lens phase term $\Phi_{lens}$ and a wavefront coding term $\Phi_{code}$ as [30,36]:(1)$$\Phi =\Phi_{lens}+\Phi_{code}=\frac{2\pi }{\lambda }f-\sqrt{x^{2}+y^{2}+f^{2}}+\Phi_{code}$$
where λ is the design wavelength, x and y are the position coordinates in the plane, and f is the focal length of the metalens. In our design, the metalens has a 1 mm-wide square aperture, a focal length f=5 mm, and a design wavelength λ=1550 nm.

For the depth-invariant EDOF metalens, the wavefront coding term of the cubic phase term [36,37,38] is used to create an accelerating Airy beam, which exhibits a highly depth-insensitive PSF with a large depth range due to its slender focus on the optical axis. The wavefront coding term of EDOF metalens is as follows:(2)$$\Phi_{code\_EDOF}=\frac{\alpha }{L^{3}}\left(x^{3}+y^{3}\right)\;\;\;\;\;\;\;\;\;\;\;\;$$
where L is half of the 1 mm aperture length and α is a constant that is the cubic phase modulation factor. Here, we set α=20π and a larger α can make the metalens have more stable depth invariance but it will increase the image noise at the same time [38]. For the depth-sensitive DH-PSF metalens, the wavefront coding term that creates a two-foci rotating beam is determined based on the spiral phase, as described by the equation below:(3)$$\Phi_{code\_DH}=\{\begin{array}{c} \left(2n-1\right)\theta ,\;\;\;\;\;\sqrt{\frac{n-1}{N}}<\rho <\sqrt{\frac{n}{N}},\;\;\;\;n=1,2,\ldots ,N\\ 0,\;\;\;\;\rho >1 \end{array}\;\;\;\;\;\;\;$$
where $\Phi_{code\_DH}$ is expressed in polar coordinates (ρ,θ) and N is the total number of Fresnel regions that can influence the rotation rate of two focal points of DH-PSF to make it sensitive to changes within the working depth [39,40] (see Appendix A). Here, we choose N=4 as an example in our design.

The bifunctional metalens consists of GST square nanorods on the SiO_2_ substrate. GST was selected because its phase-change switching speed can reach the nanosecond level or less [41] when taking two images in almost a single snapshot. Moreover, GST has the characteristics of moderate optical losses [42] and large refractive index transformation [43] in near-infrared region to well satisfy the phase changes of each unit cell between EDOF and DH-PSF metalenses. Additionally, square nanorods can provide the advantage of polarization independence (see Appendix A). The nanorods in our design have a height h=1.6 μm and period p=800 nm. Figure 1c,d present the phase and transmission as a function of side length a, ranging from 100 nm to 700 nm,calculated by rigorous coupled-wave analysis. For more details about the unit structure parameters, please refer to Appendix A. We can see that the phase cover of the GST nanorods has multiplied fourfold from the C-state ($n_{C}=2.4+0.02i$) to the A-state ($n_{A}=5.2+0.1i$) [44], which benefit from the resonance effect owing to such high index of $n_{A}$; therefore, two distinct phase profiles can be provided under the same metalens design. Although the transmission of the GST nanorods is not very high due to material absorption, which can affect the final metalens efficiency and image quality, we can use an intense light source to illuminate during imaging to help resolve this. Here, for demonstrating the principle, we assumed that the amplitude of diffraction plane equals one during the process of the image reconstruction. The corresponding nanorods with the adaptable phase at two different states were then assigned to every position of the metalens [45], and the EDOF metalens was designed in the A-state of GST while the DH-PSF metalens was designed at C-state. (See Methods and Appendix A for more details on the design of the metalens.)

The metalens includes 1250×1250 GST nanorods in total. Figure 2a shows the top view of the partial layout of the metalens. (See Appendix A for the dimensional distribution of the metalens.) It is difficult to perform full-wave simulations for the metalens because its aperture is much greater than its wavelength. Therefore, we selected one row of the nanorods from the metalens with the typical dimensions to simulate their near-field distribution using the finite difference time domain (FDTD) in order to observe their coupling strength. Figure 2b shows the |H| distributions in ten adjacent nanorods illuminated at TM polarization. It can be observed that the fields are mainly concentrated inside the individual nanorods for both states due to the high index of GST, which has few interactions with the neighboring nanorods [46,47,48]. So, the light scattering of the metalens is dominated by the individual responses of each nanorod, which is mainly a local effect [37,49]. Figure 2c,d demonstrate the generated phase plane of two metalenses by ignoring the interaction between the structural coupling, similar to the method of treatment adopted in many articles for designing large-aperture metalenses [30,50].

Figure 3a,b show the simulated PSFs in the far field of the designed bifunctional metalens using the direct integration method. In Figure 3a, it can be seen that the PSFs of the EDOF metalens varied minimally with the point light source’s shift to different depths along the optical axis (the similarities between three EDOF-PSFs and their modulation transfer function have been evaluated in Appendix A). Figure 3a also shows that the EFOD metalens’ PSF is not a standard focus point, resulting in blurry images. By implementing a single-shot measurement and subsequent deconvolution, a reconstructed clear image over a wide depth range can be achieved. This design achieves a resolution of 625 lp/mm with a highly depth-invariant modulation transfer function when compared to a design without the wavefront coding term added (see Appendix A). In addition, the depth-sensitive DH-PSF metalens creates a spatially rotating two-foci PSF that strongly depends on the depth change, as shown in Figure 3b. Therefore, we can calculate the resultant DH-PSFs of the objects through deconvolution. The depth distance of the objects can then be determined by examining the rotation angles of DH-PSFs. In Figure 3c, our DH-PSF metalens exhibits a large rotation angle variation (80.5°) over the measured depth range (3.5–13.5 cm), and the actual rotation angle of double-helix foci as a function of the object depth can be aligned with the theoretical curve calculated from the phase distribution of the metalens.

### 2.2. Depth Imaging Simulation of the Bifunctional Metalens

The simulation of depth imaging of a single-object scene with different depths was carried out first for our bifunctional metalens. The incident light with the information of a university emblem pattern was illuminated and passed through the metalens, and two blurry images were captured in the focal plane (receiving screen) by convoluting the incident scene with the EDOF-PSF and DH-PSF, as shown in Figure 4a. Then, the scene was reconstructed by applying a variation-regularized deconvolution algorithm to the cubic image of EDOF metalens to achieve a clear emblem image. For the depth estimation, we should examine the DH-PSF of the emblem, which can be calculated through deconvolution on the other helical image of DH-PSF metalens. Figure 4b shows the calculated DH-PSF of the emblem image located 7.4 cm away from the metalens. With the calculated DH-PSF, we estimated the different depths of the emblem, as shown in Figure 4c. It can be seen that our estimated depths are basically consistent with the true depth, yielding a depth precision rate of 3.42% error (see Appendix A for detailed data). Using this framework, we have reconstructed the corresponding depth map of the emblem located at five different distances (3.7 cm,5 cm,7.4 cm,9.6 cm,and 12.7 cm), as shown in Figure 4d.

In the case of the single-object scene, the accuracy of our bifunctional metalens has been demonstrated, where its scene and reconstruction process is relatively simple and the off-axis imaging aberrations had little impact on the depth precision as a little rotation offset because the object was located near the center of the field of view. Then, we implemented the simulation on the scenes of two objects with different depths located off-axis with larger angles. The actual depth of the university emblem is 4.2 cm while that of the “TJ” character is 10.4 cm. Figure 5a shows the blurred images generated by the metalens in two states. The clear image that we reconstructed, using the method described above, is shown in Figure 5b. The edge contours of the “TJ” characters are clear and have good imaging quality. In the school emblem patterns, the edges of some letters and patterns are a little blurred, so the imaging quality of more complex objects is not as high as that of simple characters. The depth estimation for the scene with multiple objects here relies on the image segmentation to manipulate two objects. The captured image of the DH-PSF metalens and the reconstructed clear images allowed us to calculate distinct DH-PSFs for each object, as shown in Figure 5c. Then, we were able to estimate the depths and establish the depth maps, as shown in Figure 5d,e (see Appendix A for detailed data). The depth estimation yields an error rate of 5.63%, which is higher than, but of a similar order, to the single-object case. This is due to the off-axis focal shift. This issue can be resolved by compensating for the rotation angle shifts using a modified reconstruction algorithm to improve the depth precision, which has been demonstrated in Ref. [30].

## 3. Discussion

We have demonstrated a new method for 3D depth imaging using an active bifunctional metalens, which can enable depth estimation in an ultracompact and coaxial optical system without having to take multiple exposures compared to various 3D depth imaging techniques existed. By integrating the imaging lens and two wavefront coding functions into a single metalens device, the size is reduced significantly. This comes at the expense of introducing a larger off-axis aberration than a refractive lens system, resulting in the focal shift and rotation offset. Performing the functionality of EDOF-PSF and DH-PSF using a single-aperture metalens not only eliminated the parallax errors induced by space multiplexing, but also avoided the requirement of scene separation previously experienced in PSF engineering methods that utilized two juxtaposed metalenses, which can further improve the field of view and depth precision of the metalens. Through the theoretical simulation of 3D depth imaging on both single-object and double-object scenes, we have demonstrated the good scene reconstruction capabilities and depth precision of our bifunctional metalens. Compared to other metasurface-based depth cameras, our design can also accommodate a wider depth detection range, which is not limited to the scope of our work. By adjusting the aperture, focal length, and phase function factors (the constant α, N etc.), a larger working depth range can be achieved to satisfy different application requirements. In addition, it should be noted that our method needs to segment the image in the image reconstruction when dealing with the multiple-object scene, which inevitably limits the transverse resolution of the depth maps. This is the shortcoming of our method when compared to conventional methods that can adapt to a denser depth scene.

The experimental feasibility of our bifunctional metalens must also be analyzed. The GST nanorods array can be fabricated through a standard electron beam lithography technique, followed by a dry etching process. It is worth mentioning that our metalens design has a large height and high aspect ratio in order to satisfy the distinct phase requirements of two states metalenses, and the nanoscale etching of GST with high aspect ratio is usually difficult. However, this issue can be resolved by process development, and has been experimentally proven before [51]. To further realize the phase-change conversion, it is feasible to add an ITO layer between the substrate and GST nanorods, whose refractive index is close to SiO_2_. As a transparent electrode, the ITO layer has sufficient conductivity to allow electrical Joule heating for GST, so the state of GST can be switched reversibly by applying an electrical current pulse through the conductive layer, as demonstrated in Ref. [52]. Therefore, our designed metalens provides the possibility for experimental demonstration in the future.

## 4. Conclusions

We have proposed an active bifunctional metalens based on phase-change material GST for 3D depth imaging. Our approach relies on a pair of complementary PSFs to generate a focused accelerating and rotating beam, which totally integrates into one metalens, enabling the image to be a compact snapshot and contain no parallax. Our approach only requires a $1\;{\mathrm{mm}}^{2}$ optical metalens without additional imaging lens and complex settings. We have also demonstrated that our method has good depth estimation precision and a good working range, with no need to capture images repeatedly. By appropriately tuning the metalens design to different focal lengths, depth ranges and sensitivity, this 3D imaging approach can have a wide range of application scenarios, such as head-mounted displays, robot vision and facial recognition.

## 5. Methods

Metalens Design: The transmission coefficients of the GST nanorods in the A-state and C-state were first extracted using a rigorous coupled wave analysis method. Using the scanning results as a database, the metalens was designed by assigning the side length for each position to satisfy the desired phase for both PSFs in different states simultaneously with the smallest phase error (see Appendix A). Then, the nanorods in the array were treated as complex amplitude pixels, forming a phase plane to perform the far-field PSF simulation based on the direct integration method using MATLAB codes.

Deconvolution: The scene reconstruction and depth estimation were completed through the following steps: (1) The scene images were convoluted with the PSFs generated by the metalenses to simulate the blurred cubic and helical images. (2) The reconstructed scenes were calculated by deconvoluting the cubic images generated by the EDOF metalens to restore the clear images using a regularization-filter based deconvolution algorithm. (3) After reconstructing the image, we segmented the image into several subregions of each object for depth estimation. We used the depth estimation algorithm based on the Wiener filter to deconvolute the helical image generated by DH-PSF metalens to obtain the estimated DH-PSF of each object. (4) The depth of the object was estimated by examining the rotation angle of the DH-PSF, comparing this against the theoretical angle-depth curve. For more information on scene reconstruction and depth estimation, please refer to Appendix A. In addition, two deconvolution procedures in our depth imaging relied on the algorithm processing time, which averaged 1.6 s and 0.5 s, using an ordinary desktop computer (CPU, Intel Core i7-11700) with the algorithm implemented in MATLAB. So, the depth imaging is not real time after completing the 3D scene reconstruction. By improving the hardware of the computer, such as the GPU, the processing speed can be improved, which would make real-time imaging feasible.

## Figures and Tables

**Figure 1 micromachines-13-00607-f001:**
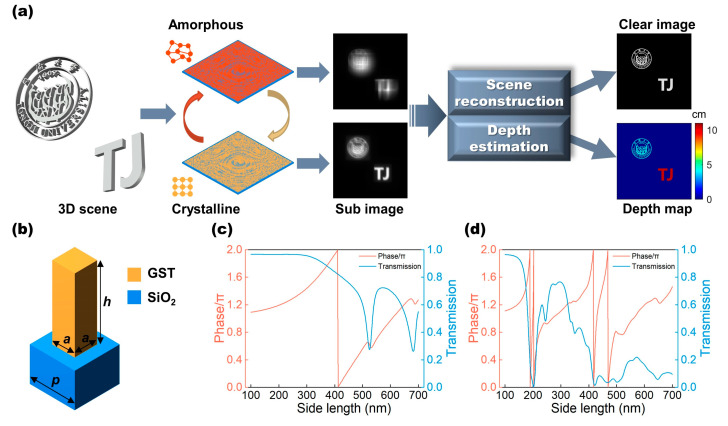
(**a**) Schematic diagram of working principle of the bifunctional metalens. (**b**) Schematic view of the unit cell, consisting of GST nanorod and silica base. The geometric parameters include side length a, period p, and height h. In (**c**,**d**), the phase and transmission of C-state and A-state GST nanorods are shown as a function of side length a, respectively.

**Figure 2 micromachines-13-00607-f002:**
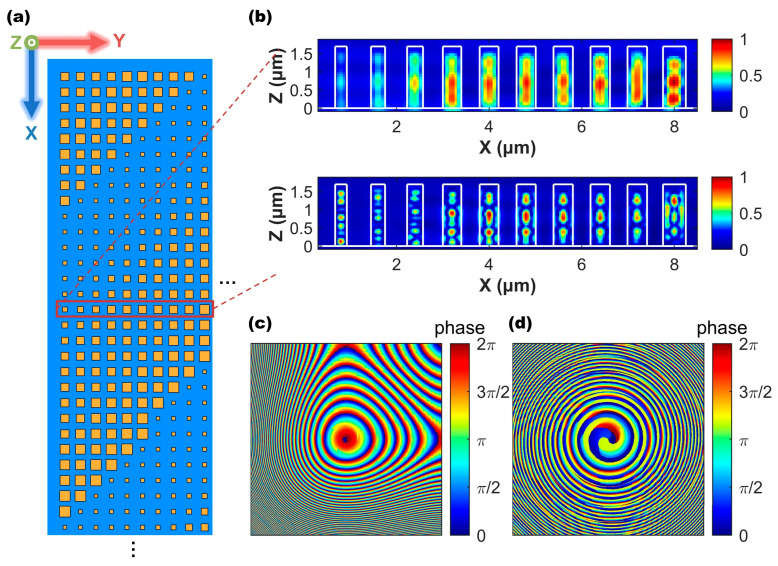
(**a**) Top view of the partial arrangement of GST nanorods in the designed metalens. (**b**) The magnetic field amplitude |H| in the ten unit structures of C-state (upper) and A-state (lower) is distributed in the X−Z plane at 1550 nm and TM polarization. Their period is 0.8 μm, and h=1.6 μm. The side lengths of these nanorods from left to right are 0.252 μm, 0.304 μm, 0.344 μm, 0.398 μm, 0.408 μm, 0.414 μm, 0.418 μm, 0.420 μm, 0.422 μm, and 0.488 μm, respectively. In (**c**,**d**), they have shown the actual phase distribution of the whole EDOF metalens in A-state and C-state, respectively. The size of the metalens is 1 mm×1 mm.

**Figure 3 micromachines-13-00607-f003:**
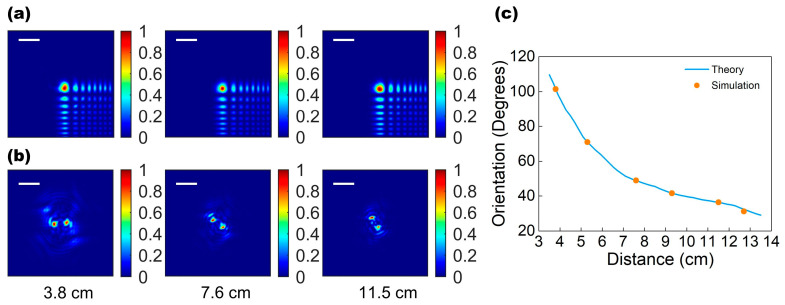
(**a**) The PSF generated by EDOF metalens at three different depths that are essentially unchanged with object depth. (**b**) The DH-PSF rotates around the center at three different depths. The depths in (**a**,**b**) are 3.8 cm, 7.6 cm, and 11.5 cm. Scale bar: 64 μm. (**c**) The theoretical relationship curve between DH-PSF rotation angle and object distance (blue line); the simulated results of rotation angle of DH-PSF varied with the object distance (orange points).

**Figure 4 micromachines-13-00607-f004:**
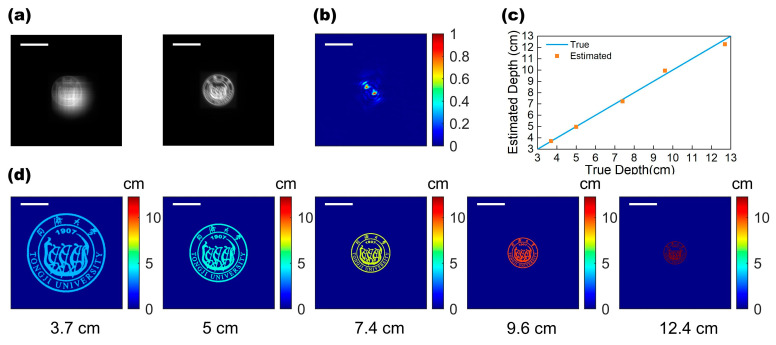
Single-object depth imaging. (**a**) Convolution blurred images (depth 7.4 cm) on simulated receiving screen, including cubic image of EDOF metalens (**left**) and helical image of DH-PSF metalens (**right**). Scale bar: 0.5 mm. (**b**) The DH-PSF was estimated using the image in (**a**). Scale bar: 0.1 mm. (**c**) The estimated object depths (orange points) are compared with the actual depth (blue line). (**d**) Clear reconstructed depth images of five different distances with the same color bar. Scale bar: 0.385 mm.

**Figure 5 micromachines-13-00607-f005:**
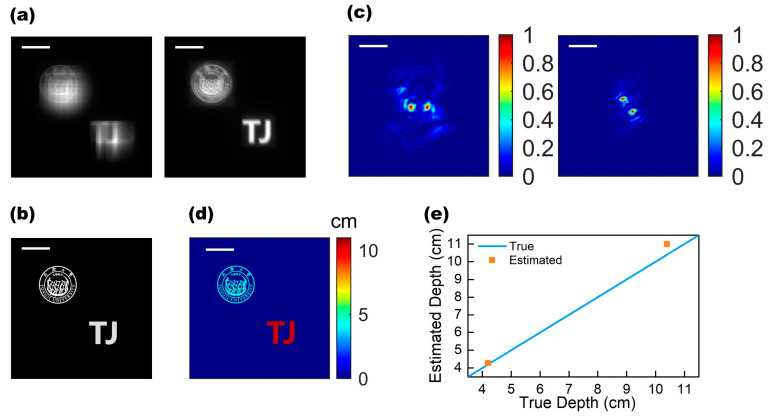
Dual-object depth imaging. (**a**) Convolution blurred images of two objects with different depths. EDOF metalens image (**left**) and DH-PSF metalens image (**right**). Scale bar: 0.4 mm. (**b**) Reconstructed images of two objects. Scale bar: 0.4 mm. (**d**) Depth map of two objects (near badge and far “TJ” character). Scale bar: 0.4 mm. (**c**) The DH-PSFs of the two objects were estimated using the image in (**a**). DH-PSF of the school emblem (**left**) and DH-PSF of the “TJ” character (**right**). Scale bar: 64 μm. (**d**) Depth map of two objects (near school emblem and far “TJ” character). Scale bar: 0.4 mm. (**e**) The comparison between the estimated depth of the two objects (orange points) and the actual depth (blue line).

## Data Availability

Not applicable.

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
