# Peer review of "Reconfigurable Metalens with Phase-Change Switching between Beam Acceleration and Rotation for 3D Depth Imaging"

_micromachines, 2022, doi:10.3390/mi13040607_

Round 1

Reviewer 1 Report

Overall Comment:
The article requires major revisions, and the current version is not suitable for publication.
1. This is a simulation paper with no experimental validation. Moreover, the specific simulation process is not clearly described.

I provide some technical points that authors should check:

Technical Comment:
1. The principle part needs to be rewritten. Due to the logic problem of the article, the introduction of metalense principle is full of too many assumed descriptions. I suggest that the author clarify the logic and describe why we do it, rather than just do it.
2. Although this is a simulation article without experiment, it is still necessary to describe and analyze the feasibility of the experiment.

3.  A lot of information needed to replicate the results is missing in the manuscript:
(a) What are the specific parameters of the nine GST structures?
(b)  From "The far-field image shows that the PSF generated by the EDOF metalens (Fig. 3a) is almost unchanged with the change of object distance.",in fact, I can't see the specific supporting data of this sentence. I don't even know the exact size of all the images related to imaging in the article.
(c) "but actually we only selected the diagonal line structures (white straight line in Fig. 1c,d) where has a = b= l to eliminate the polarization dependence", since polarization dependence is not required, why provide useless information that a is not equal to b in Fig. 1(C) and (D)?
(d) Since the design of the article is for imaging, the efficiency problem needs to be discussed and analyzed. There is no place in the full text to provide the efficiency of the design metalens, and even the efficiency of a single structure is not provided in the structural unit database of part 2.1.
(e) All pictures need to be modified, including using clearer pictures; For the color bar describing the phase, please refer to the format of other articles; Unify the font size of numbers, letters, etc. in the picture;Delete pictures with duplicate information (Fig. 4 (e) or (d), etc), and provide pictures with more new information (Fig. 5 (e)), etc.
(f) Supplement the performance analysis of the metalens, such as resolution, imaging quality under the design limit, etc.
(g) The depth imaging process in this paper depends heavily on algorithm processing, so what is the specific calculation time? CPU based or GPU based? Are there any algorithmic innovations? What are the pixels of the processed image?
(h) The design goal of polarization insensitive is proposed in the design of the article, but it is not clearly discussed whether the design goal is achieved in the subsequent part of the manuscript.

I would have overlooked these faults if this was an experimental paper but since it is primarily a simulation paper, these things become very important. Otherwise a slight mismatch here and there, experimental results won't match simulations.

7. "which will reduce the depletion efficiency". How are the authors characterizing this depletion efficiency?

8. Finally, it would be good if the authors can at least do a 2D image simulation with the PSFs to showcase the imaging performance. It is hard to predict anything from simple PSFs. Moreover, what are the Strehl ratio and the MTF for the focussing metalenses? 

Reviewer 2 Report

In this work, a metalens is modeled, which, at different refractive indices of its structural elements (nanorods), forms two different (arbitrarily specified) light beams (an accelerating beam and a rotating two-leaf beam). One metalens with different refractive indices (the ratio of these indices nC/nA=1.7) has a given ratio of refractive indices in each atom of the metalens, and hence the given phase ratio ФС/ФA=1.7. But the value of the phase ratio at the point does not have to be the same, it may differ from the value of 1.7: Ф(EDOF)/Ф(DH-PSF) < > 1.7. Therefore, one metalens, which, at different refractive indices, forms different arbitrarily given diffraction patterns, cannot be calculated. In this work, the authors did not show that the same metalens structure forms different (arbitrarily specified) light fields. In the work, there is not even a drawing of the entire surface of this metalens, but only a small fragment of it (Fig. 1a). Therefore, the fact that figures 3a and 3b are formed by the same metalens raises serious doubts. In order not to mislead the reader, I do not recommend publishing this work.

Round 2

Reviewer 1 Report

The author answered many of my questions, supplemented much data, and adjusted the structure of the article. The whole article is well organized at present. When the author answers my new questions, I recommend publication in Micromachines.

The first paragraph of the introduction, more relevant references should be added and the novelty of this manuscript should be explicitly addressed. (1) Spherical Aberration-Corrected Metalens for Polarization Multiplexed Imaging, Nanomaterials 2021, 11, 2774 (2) Achromatic terahertz Airy beam generation with dielectric metasurfaces. Nanophotonics, 10, no. 3, 2021, pp. 1123-1131. (3) Broadband achromatic metalens in terahertz regime. Science Bulletin, Volume 64, Issue 20, 2019, Pages 1525-1531.

Reviewer 2 Report

The authors have substantially revised this work, and therefore it can be published